# Exploring discrepancies between in vivo and simulated correction in 3D-planned distal radius osteotomies: the influence of biological factors

Emilia Gryska[1,2]*, Katleen Libberecht[1,2], Per Fredrikson[1,2], Charlotte Stor Swinkels[1,2,3], Peter Axelsson[1,2], Anders Björkman[1,2]

1 Department of Hand Surgery, Sahlgrenska University Hospital, Mölndal, Sweden, 2 Institute of Clinical Sciences, Sahlgrenska Academy, University of Gothenburg, Gothenburg, Sweden, 3 Department of Medical Physics and Biomedical Engineering, Sahlgrenska University Hospital, Gothenburg, Sweden

* emilia.gryska@gu.se

## Abstract

Three-dimensional virtual surgical planning (3D VSP) with patient-specific surgical guides (PSSGs) can improve the accuracy of corrective distal radius osteotomy. Simulation on 3D-printed bone models allows controlled evaluation of planned corrections, but such models do not reproduce biological factors, including bone quality and soft-tissue tension. The extent to which simulated corrections reflect in vivo surgical results remains unclear. This exploratory study included 13 patients who underwent corrective distal radius osteotomy using 3D VSP and PSSGs. Simulated osteotomies were performed on patient-specific 3D-printed bone models. Residual correction errors in volar tilt (VT), radial inclination (RI), and radial length (RL) were calculated relative to the 3D VSP for both in vivo and simulated corrections. Discrepancies between in vivo and simulated corrections were assessed, and their associations with age, bone mineral density estimated from Hounsfield units, and planned maximum distraction were explored using Spearman's rank correlation. Residual correction errors in simulated osteotomies were generally comparable to those observed in vivo. The largest differences between simulated and in vivo corrections were observed for VT (3.2), although the mean difference remained within clinically acceptable limits. A positive correlation was found between RL discrepancy and planned maximum distraction ($r_s = 0.71$, 95% CI: 0.27–0.91, $p = 0.01$), and a negative correlation was found between RL discrepancy and bone mineral density ($r_s = -0.64$, 95% CI: −0.88 − −0.13, $p = 0.02$). No correlations were identified for VT or RI. Simulated distal radius osteotomies on 3D-printed bone models produced correction errors broadly comparable to those achieved in vivo. However, small but systematic discrepancies in VT and RL suggest that biological factors may influence the relation between simulated and clinical outcomes. These findings support the use of simulation models for comparative studies while highlighting the need for cautious interpretation of simulation-based results.

**Data availability statement:** All relevant data are within the manuscript and its Supporting Information files.

**Funding:** AB received grants from the Swedish state under the agreement between the Swedish government and the county councils, the ALF-agreement (ALFGBG-966260). The funders had no role in study design, data collection and analysis, decision to publish, or preparation of the manuscript.

**Competing interests:** The authors have declared that no competing interests exist.

## Introduction

Several previous studies have shown that three-dimensional virtual surgical planning (3D VSP) combined with patient-specific surgical guides (PSSGs) can improve the accuracy of corrective osteotomies of the distal radius compared with conventional free-hand techniques [1,2]. Nevertheless, residual errors relative to the virtual plan remain common [3–5]. Surgical accuracy may be influenced by technical factors such as guide-to-bone fit [6], the design of the PSSG [3,5], and the resolution of the CT images used for planning [7]. Evaluating the relative contribution of such factors is difficult in clinical studies because each patient presents with a unique combination of anatomy, bone quality, and deformity. This inter-patient variability makes it challenging to isolate the effect of a single technical factor, for example, two guide designs, without confounding from patient-specific differences. Patient-specific 3D-printed bone models, which have been shown to reproduce bony surface anatomy accurately [8], provide a potential solution. Such models make it possible to compare different surgical approaches under controlled conditions for the same deformity, which is difficult to achieve in clinical cohorts [6,9,10].

However, simulated osteotomies on 3D-printed models do not reproduce important biological conditions present during actual surgery, including bone quality and soft-tissue tension. As a result, the correction achieved in simulation may differ from the correction achieved in vivo. The magnitude of this difference, and whether it is associated with factors related to bone density or soft-tissue tension, has not been established. This limits the external validity of simulation-based studies using 3D-printed models.

The aim of this exploratory study was to assess discrepancies between in vivo and simulated correction in 3D-planned distal radius osteotomy and to examine whether these discrepancies were associated with soft tissue- and bone density-related factors. Specifically, we explored the associations of age, bone mineral density, and planned distraction with discrepancies in volar tilt (VT), radial inclination (RI), and radial length (RL) between the simulated and in vivo surgery. We also examined the differences and agreement in residual errors between in vivo and simulated corrections.

## Materials and methods

### Study design, and participant cohort

This retrospective observational study represents a secondary analysis of prospectively recruited consecutive patients who underwent corrective osteotomy for extra-articular distal radius malunion at Sahlgrenska University Hospital between 13 July 2021 and 13 July 2023. The cohort was originally enrolled as part of a previously published prospective study (Stor Swinkels et al. [10]).

Sixteen eligible patients were identified during the inclusion period. Three were excluded from the present analysis (one withdrew consent and two had insufficient postoperative CT image quality), leaving thirteen patients for final evaluation. The study was approved by the Swedish Ethics Review Board (2021−01974). All

patients gave written informed consent after receiving oral and written information about the study. The patients' data were accessed between September 15th, 2023, and June 26th, 2024. The authors could identify participants during and after data collection.

## Image and virtual model acquisition

Preoperative and postoperative CT scans were acquired using a GE Discovery CT750 HD scanner (GE Healthcare, Milwaukee, WI, USA) with a pixel size of 0.39 × 0.39 mm, a slice thickness of 0.625 mm, a tube voltage of 100 kVp, and a tube current of 100 mA. Postoperative CT scans were obtained within one week after surgery to evaluate immediate correction accuracy relative to the 3D virtual surgical plan; long-term subsidence or implant migration was not assessed. The radius bone was segmented from CT datasets using Mimics software (Materialise NV, Leuven, Belgium), and three-dimensional virtual bone models were generated. The preoperative model served as the basis for 3D VSP.

## Biological and clinical variables

The following biological and clinical variables were analyzed: age at the time of surgery, bone mineral density (BMD), and planned maximum distraction. As dual-energy X-ray absorptiometry (DEXA) measurements were not available for this cohort, BMD was estimated from preoperative CT scans using Hounsfield unit (HU) measurements. We applied the method described by Wagner et al. [11], who demonstrated that HU values obtained from the distal ulna correlate with bone mineral density and may serve as a surrogate measure in clinical studies. The distal ulna was selected because it was consistently included in all preoperative CT scans and is anatomically adjacent to the surgical site. No metal artifacts were present in the region of interest. The mean HU value from three consecutive coronal slices was used for analysis (Fig 1). For CTs at 120 kVp, HU values below 146 indicate osteopenia, and below 121 indicate osteoporosis [11].

The planned maximum distraction of the radius was defined as the largest dorsal osteotomy gap measured at the dorsal–radial aspect of the osteotomy in the lateral column (radial styloid region) [12] on the 3D VSP.

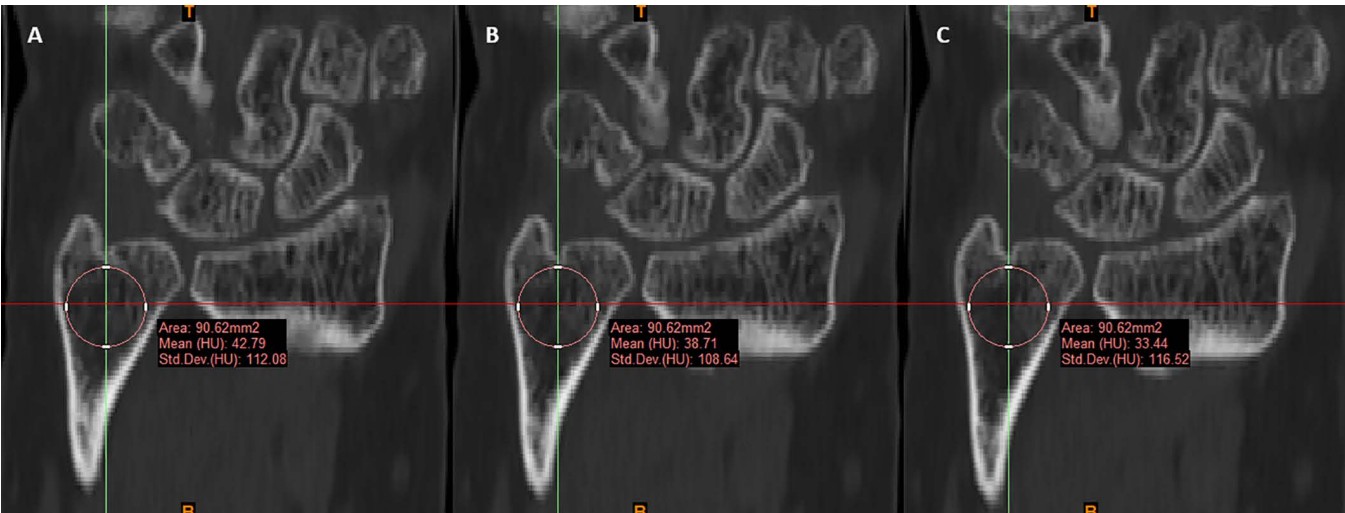

**Fig 1. Bone mineral density measurement according to Wagner et al. [11].** A circular region of interest was placed within the cancellous bone of the ulnar head. Measurements were obtained on three consecutive coronal CT slices A-C at the level of the distal radioulnar joint and averaged for analysis.

## Assessment of in vivo (IV) correction accuracy

Osteotomies were fixated using a Synthes VA-LCP two-column volar distal radius plate in 11 patients and a Stryker Variax volar distal radius plate in two patients. The type of plate was dictated by the surgeon's preference, the availability of STL files, and the plate fit on the bone during virtual planning. The decision to release the brachioradialis was made at the operating surgeon's discretion but was not documented consistently. The accuracy of the correction was evaluated against 3D VSP (Fig 2A) using postoperative CT-derived 3D models (IV models, Fig 2B). Each IV model was aligned to its corresponding 3D VSP model in 3-matic (Materialise NV, Leuven, Belgium). The rigid transformation matrix describing translation and rotation between the IV model and the planned model was extracted, and residual correction errors in volar tilt (VT, °), radial inclination (RI, °), and radial length (RL, mm) were calculated relative to the 3D VSP.

## Assessment of simulated (SIM) correction accuracy

To assess simulated correction accuracy, the preoperative radius models used for the 3D VSP were 3D-printed for each patient. A surgeon experienced with the 3D VSP technique (PA) performed the planned surgical procedure on the 3D-printed radiuses with a duplicate set of PSSGs and osteotomy plates identical to those used in vivo. After simulated correction, the 3D-printed models were scanned (Einscan SP, Shining 3D® Tech. Co., Hangzhou, China) to create a virtual simulation (SIM) model of each radius (Fig 2C). These SIM models were aligned to their respective 3D VSP models, and residual errors in VT (°), RI (°), and RL (°) were calculated using the same method as for the IV models.

## Assessment of discrepancy between IV and SIM

The discrepancy between IV and SIM corrections was calculated as the relative rigid transformation between the two models. Specifically, the transformation mapping SIM to IV was computed as:

$$T_{SIM \rightarrow IV} = T^{-1}_{IV \rightarrow 3D\ VSP} \cdot T_{SIM \rightarrow 3D\ VSP},$$

where $T_{IV \rightarrow 3D\ VSP}$ and $T_{SIM \rightarrow 3D\ VSP}$ represent the rigid transformations of the IV and SIM models relative to the 3D VSP, respectively. The inverse of the IV transformation was used to express the SIM model in the IV coordinate frame. From the resulting relative transformation matrix, differences in VT (°), RI (°), and RL (mm) between IV and SIM were calculated.

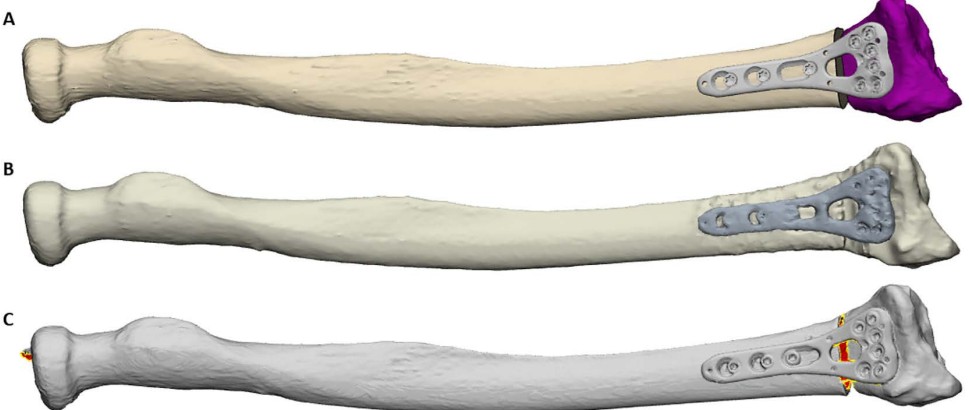

**Fig 2. Representative example of the virtual models used for analysis in one patient.** (A) Preoperative 3D virtual surgical plan. (B) Postoperative model derived from CT imaging after surgery (in vivo result). (C) Model obtained from the simulated corrective procedure performed on the 3D-printed bone (simulation result).

## Statistical analysis

Scatter plots were used to explore associations between IV–SIM discrepancies in VT (°), RI (°), RL (mm), and bone mineral density, planned maximum distraction, and age. Spearman's rank correlation coefficient ($r_s$) and corresponding 95% confidence intervals (CI) were calculated for each pair of variables, as this method is robust to small sample sizes and does not assume normality. Given the exploratory nature of the study, p-values were reported alongside effect sizes for completeness. All associations should be interpreted as hypothesis-generating. Residual errors in VT, RI, and RL for the in-vivo and simulated surgeries were reported. Residual error distributions for SIM and IV corrections were compared using boxplots, and the agreement between SIM and IV was assessed using Bland–Altman plots. Bias and limits of agreement in VT and RL analysis were compared to values below which the errors are considered clinically irrelevant: 5° for VT and 2 mm for RL [13–16].

## Results

Thirteen patients were included in the final analysis. Median age was 61 years (range 21–77). Mean bone mineral density was 444 HU (range 6–190). Planned maximum distraction ranged from 4.5 to 13.7 mm.

The correlation analysis (Fig 3) examined the association between IV-SIM discrepancies expressed as VT, RI, and RL, and age, BMD, and maximum planned distraction. No correlations were found for VT or RI. For RL, a positive correlation between the IV-SIM discrepancy and the maximum distraction was observed ($r_s = 0.71$, 95% CI: 0.27–0.91, p = 0.01). In addition, a negative correlation was observed between RL and BMD ($r_s = -0.64$, 95% CI: −0.88 − −0.13, p = 0.02).

The mean (standard deviation, SD) residual errors after in vivo surgery were −4.6° (3.1) for VT, −1.7° (2.6) for RI, and −0.4 mm (0.9) for RL. The corresponding residual errors after the simulated surgery were −1.5° (2.5), −1.3° (1.6), and −0.7 mm (1.0), respectively.

Fig 4 illustrates the distribution of the residual errors in VT, RI, and RL for the IV and SIM models. The largest difference between groups was observed for VT, with greater residual errors in IV than in SIM. For RI, the median error was slightly larger in SIM, whereas the variability was greater in IV. For RL, the median errors were similar between groups, although the SIM group showed a wider interquartile range.

Bland-Altman agreement analysis (Fig 5) showed the largest bias (3.2) for VT errors. Given that the majority of the residual VT errors were negative, the bias indicates that IV residual errors in VT are systematically larger than in SIM. The limits of agreement were −2.7 and 9.1. The upper limit of agreement exceeds the clinically relevant threshold of 5°, which has been reported as the minimum clinically relevant change in volar tilt [13–16]. We attribute this to one patient whose VT difference between IV and SIM was > 10°. The residual VT error in IV for that participant was −8.0°, while in SIM − 3.0°. The IV error could be attributed to bone density and soft tissue tension, as the patient had low BMD (11 HU and osteopenia noted in the patient's journal) and the planned maximum distraction was large: 13.7 mm. For RI and RL, the bias was smaller: 0.4° and −0.39 mm, respectively. For RI, the limits of agreement were −4.4° to 5.2°, and for the RL, −2.3 mm to 1.8 mm. Thus, the lower limit of agreement for RL slightly exceeded the clinically relevant difference of 2 mm [13–16]. Again, this is due to differences in one patient who had an IV residual RL error of 2 mm, while the SIM residual error was −0.5 mm. This patient had high BMD (158 HU), and the source of the residual error is not obvious.

## Discussion

In this exploratory study, we investigated whether soft-tissue tension and bone density are associated with discrepancies between in vivo (IV) and simulated (SIM) distal radius osteotomies performed with PSSGs. We also assessed the differences and agreement between the residual errors of IV and SIM osteotomies. There was a systematic difference in residual error between IV and SIM for volar tilt (VT), while the mean difference (SIM–IV = 3.2) remained within a clinically acceptable range. Discrepancy between IV and SIM in RL correlated with planned maximum distraction ($r_s = 0.71$,

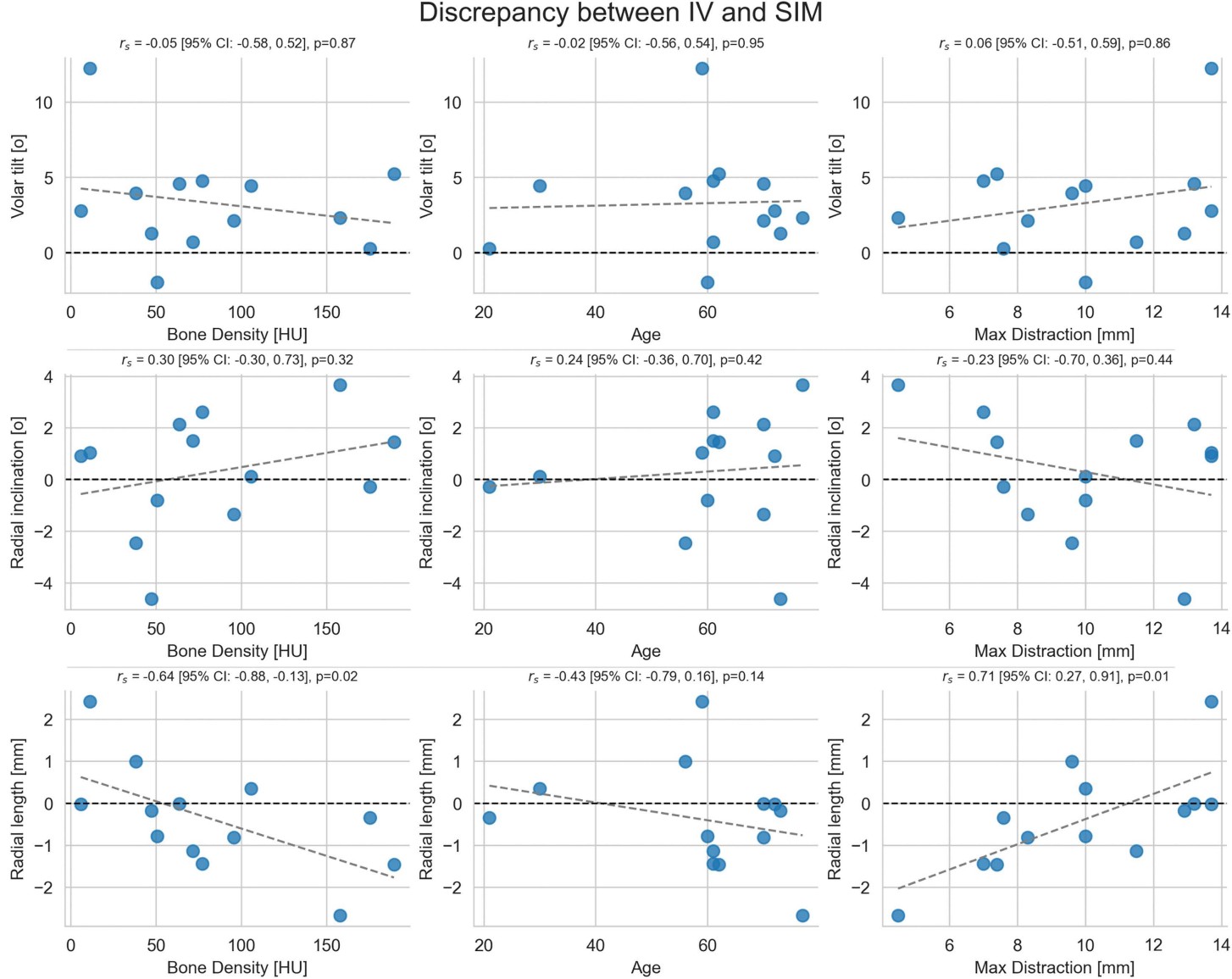

**Fig 3. Scatterplot of the discrepancy between in vivo (IV) correction and simulated (SIM) correction in VT, RI, and RL plotted against the bone mineral density expressed as average HU value in the cancellous distal ulna at the distal radioulnar joint; age at the time of surgery; and maximum distraction.** Regression lines are shown for visual guidance only. Statistical associations were evaluated using Spearman's rank correlation coefficient ($r_s$). 95% CIs and p-values are given for completeness.

95% CI: 0.27–0.91) and BMD ($r_s = -0.64$, 95% CI: $-0.8 - -0.13$), but the wide 95% CIs reflect the limited sample size and indicate that these estimates should be interpreted with caution.

Overall, these findings suggest that simulated osteotomies using PSSGs on 3D-printed bone models produce results broadly comparable to those achieved in vivo. This supports the use of 3D-printed models as a platform for investigating factors that are difficult to study directly in patients, such as evaluating alternative PSSG designs [6,9,10]. In this study, the simplified environment was intended to isolate the mechanical accuracy of the surgical plan and guide system from

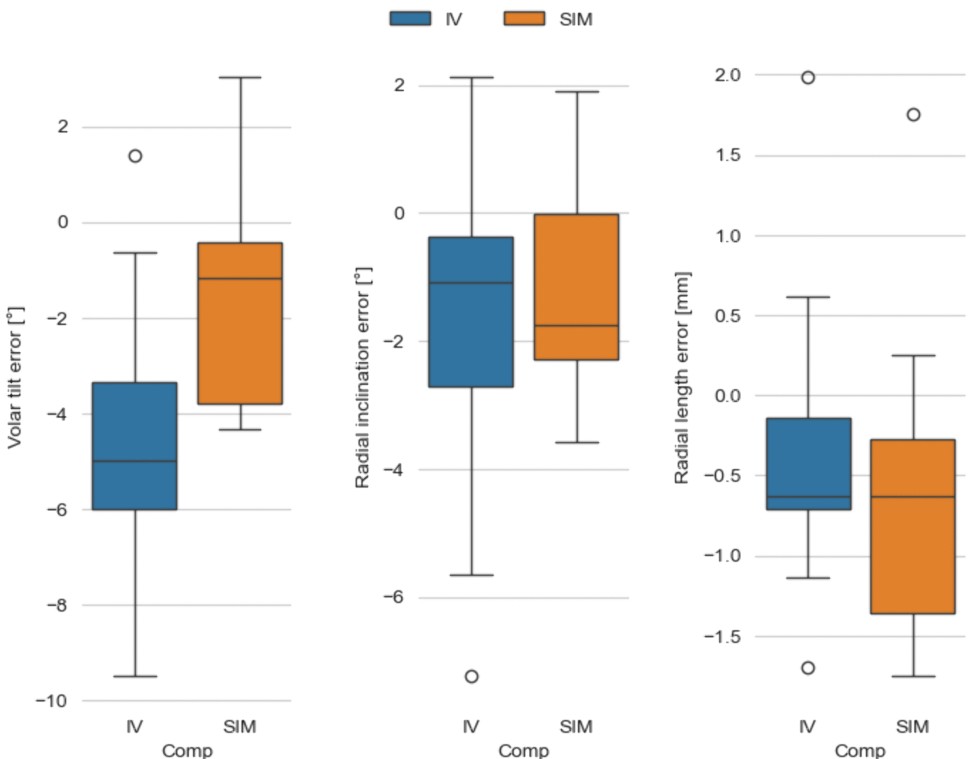

**Fig 4. Distribution of residual errors in volar tilt, radial inclination, and radial length for in vivo (IV) and simulated (SIM) corrections relative to the virtual surgical plan.** Boxes represent the interquartile range with the median indicated by the horizontal line. Whiskers indicate the range excluding outliers, which are shown as individual points.

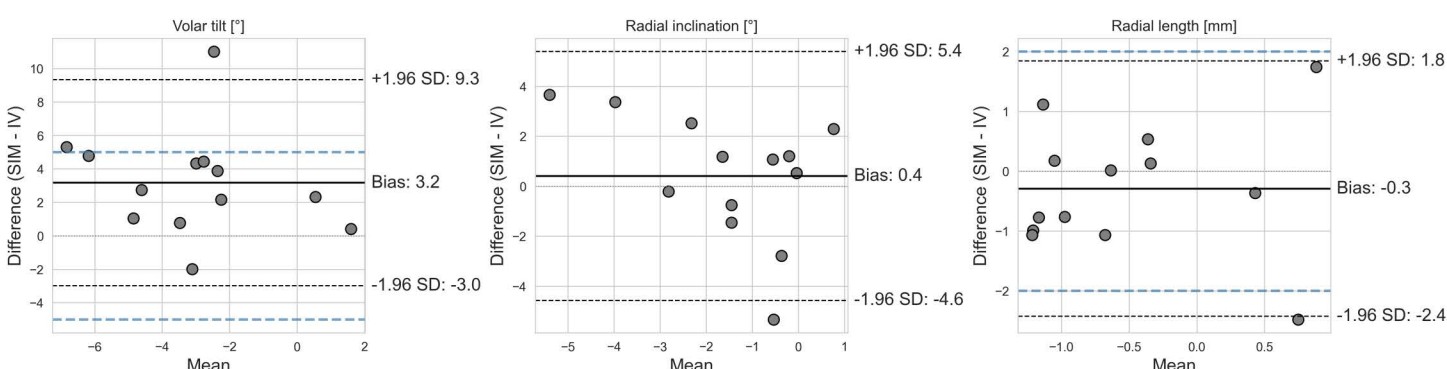

**Fig 5. Bland-Altman plot of the difference in residual errors between the simulated (SIM) surgery on 3D-printed models and in vivo (IV) post-operative models (SIM –IV).** Limits of agreement are indicated by a black, dashed line. Clinically relevant differences for volar tilt, and radial length are indicated by a blue dashed line.

factors such as bone density and soft-tissue tension, which may contribute to discrepancies between simulated and in vivo outcomes.

The observed IV-SIM discrepancies in both RL and VT may reflect intraoperative factors not captured in the simulation. Both RL and VT restoration require distraction of the distal fragment and may therefore be sensitive to soft-tissue tension and fixation stability. The systematic difference observed in VT suggests that intraoperative factors such as bone settling due to plate offset from the bone [5] are at play in vivo but not in SIM. Importantly, however, the interquartile range of the observed VT differences remained within clinically acceptable limits in our cohort [13–16]. Correlations were observed between RL discrepancies and both planned maximum distraction and BMD, though these should be interpreted with caution, given the wide confidence intervals and the interdependence between RL and VT, where loss of VT correction will also affect RL.

Although simulated procedures eliminate biological variability, they are not entirely free of technical sources of error. Dimensional deviations may arise from printer resolution limits or material shrinkage, while surface irregularities of the printed models may affect the fit of the PSSGs. In addition, drilling and fixation of guide wires during simulated procedures may cause localized material deformation, which could affect the accuracy of subsequent guide fixation.

While the present study provides important insights, several limitations should be considered. The major limitation of this study is its small cohort, which limits the generalisability of the findings. Furthermore, this study did not systematically analyse other contributors to the residual errors, like plate positioning errors, bone settling, or a mismatch between the shape of the volar plates and the slope of the volar surface in the corrected position of the distal radius [17]. A weakness of the study is that no measurements of the screw positioning relative to the subchondral plate were included, which may influence the absolute in vivo correction outcome. However, meaningfully assessing screw positioning relative to the subchondral bone in a simulated setting is not feasible because the 3D-printed models cannot replicate bone properties. Furthermore, since the planning was done in a standardised manner, there was little variation in screw positioning between cases. We therefore do not expect to find a significant correlation with the final result in this small, homogeneous sample. We also cannot directly use the HU threshold for osteoporosis proposed by Wagner et al. because of differences in the acquisition parameters [11]. Their HU threshold for osteopenia was 146 HU, and for osteoporosis 121 HU. Our acquisition protocol used lower peak kilovoltage (100kV vs 120kV); thus, the thresholds are likely different in our cohort [18]. Implant type was not included as a variable, as identical plates were used in the corresponding in vivo and simulated procedures. Finally, the release of the brachioradialis tendon, which may influence volar tilt correction but cannot be simulated in 3D-printed models, was not included in the analysis due to variability in operative documentation. Given the exploratory nature of the study and the number of correlations examined, our findings should be interpreted cautiously and considered hypothesis-generating. Simulation using patient-specific 3D-printed models enables controlled evaluation of surgical techniques while reducing patient variability. However, these models do not replicate biological conditions such as bone quality or soft-tissue tension, and discrepancies from in vivo outcomes are therefore expected. Quantifying these differences is important for clinical interpretation of simulation results. Future predictive models incorporating patient-specific factors may help improve translation to clinical practice.

Establishing such predictive models will require studies with larger cohorts and a wider range of bone mineral density to validate the observed effects. Additional factors, such as plate positioning, screw distance to the subchondral bone, and plate type, should also be evaluated.

Future work should also assess guide placement errors and their impact on distal radius correction, both in simulated and in vivo settings. While simulation using 3D-printed models is a valuable research tool, a better understanding of these factors is necessary to optimise its clinical applicability.

## Conclusions

This exploratory study evaluated discrepancies between simulated and in vivo distal radius osteotomies performed using patient-specific surgical guides and examined whether these discrepancies were associated with bone mineral density

and soft-tissue–related factors. Simulated osteotomies on 3D-printed bone models produced residual correction errors broadly comparable to those observed in vivo. However, correlations or systematic discrepancies between simulated and in vivo results in radial length and volar tilt were found, suggesting that biological factors may, as expected, influence surgical outcomes. These findings support the use of simulation models for comparative studies while emphasizing the need to consider biological factors when interpreting results from simulation-based evaluations.

## Supporting information

**S1 Data. Data generated for this study; all analyses are based on the data included in the file.**
(XLSX)

## Author contributions

**Conceptualization:** Emilia Gryska, Katleen Libberecht, Per Fredrikson.

**Data curation:** Emilia Gryska, Katleen Libberecht, Charlotte Stor Swinkels, Peter Axelsson.

**Formal analysis:** Emilia Gryska, Charlotte Stor Swinkels, Peter Axelsson.

**Funding acquisition:** Anders Björkman.

**Investigation:** Emilia Gryska.

**Methodology:** Emilia Gryska, Katleen Libberecht, Per Fredrikson.

**Supervision:** Anders Björkman.

**Visualization:** Emilia Gryska.

**Writing – original draft:** Emilia Gryska.

**Writing – review & editing:** Katleen Libberecht, Per Fredrikson, Charlotte Stor Swinkels, Peter Axelsson, Anders Björkman.

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
