## [Decision Letter · Decision Letter 0]

12 Feb 2026

PONE-D-25-62620Influence of bone mineral density and planned distraction on residual errors in 3D-planned distal radius osteotomies: an exploratory analysisPLOS One

Dear Dr. Gryska,

Thank you for submitting your manuscript to PLOS ONE. After careful consideration, we feel that it has merit but does not fully meet PLOS ONE’s publication criteria as it currently stands. Therefore, we invite you to submit a revised version of the manuscript that addresses the points raised during the review process.

We look forward to receiving your revised manuscript.

Kind regards,

Furqan A. Shah

Academic Editor

PLOS One

Journal Requirements:

Reviewers' comments:

Reviewer's Responses to Questions

**Comments to the Author**

1. Is the manuscript technically sound, and do the data support the conclusions?

Reviewer #1: Yes

Reviewer #2: Yes

Reviewer #3: Partly

2. Has the statistical analysis been performed appropriately and rigorously? 

Reviewer #1: Yes

Reviewer #2: Yes

Reviewer #3: Yes

3. Have the authors made all data underlying the findings in their manuscript fully available?

Reviewer #1: Yes

Reviewer #2: Yes

Reviewer #3: Yes

4. Is the manuscript presented in an intelligible fashion and written in standard English?

Reviewer #1: Yes

Reviewer #2: Yes

Reviewer #3: Yes

5. Review Comments to the Author

Reviewer #1: The manuscript investigates how bone mineral density and planned maximum distraction influence residual correction errors in 3D-planned distal radius osteotomies performed with 3D virtual surgical planning (3D VSP) and patient-specific surgical guides (PSSGs). The topic is clinically relevant and contributes to imporving accuracy in corrective osteotomy.

However, some issue should be considered and addressed:

The objective is not clearly stated and there is no hypothesis.

The term “clinical parameters” is not optimal and it is not clear from the outset that a comparison will be made. It could be replaced by “influence of soft tissue tension and bone density.”

The hypothesis could refer specifically to the comparison between actual surgery and surgery performed on 3D models.

In terms of methodology, this would also mean better identification and description of the two groups: “actual surgery” and “surgery on 3D models” in a dedicated chapter.

The rationale for comparing simulated and actual surgeries should be more explicitly linked to the clinical question.

Methods

The methods should be divided in different chapters with subtitle.

Line 20: It would be clearer and more illustrative to include a figure showing how the measurements were obtained

De la même façon, il manque des images du modèle 3D utilisé et avec la reduction.

Line 2 (p5): why “consecutive”? So specify the design of the study anad the period of the inclusion

Line 16 (p5): wagner method, why did the author choose this method ?

Line 1 (p6): specify the units (VT, RI, RL…)

Line 1 (p7): difference was expressed in SIM-TRU or TRU-SIM ?

Results

Line 1-3 (p9): Add a short explanation of the clinical relevance thresholds

Line 8-14 (p9): Provide more interpretation, especially since some limits of agreement exceed clinically acceptable bounds.

Discussion

Line 25 (p11): “3D-printing errors affect the guide and the bone model” please specify

Conclusion: too long, should be only related to the findings

Remove : “Understanding how bone quality influences the surgical outcomes will enable the development of predictive models for correction loss, allowing for improved surgical planning and optimized 3D techniques. Further studies are needed to understand errors in simulated surgeries to make it a robust research method in 3D VSP and PSSG”

Reviewer #2: The authors have provided a manuscript of their study analyzing the influence of bone mineral density and planned distraction on distal radius osteotomies and the accuracy of simulated corrections.

The topic is highly relevant.

The methodology is clearly described.

The results are presented clearly and the figures are relevant and contribute meaningfully to the manuscript.

Reviewer #3: Thank you for letting me review this paper. I have some general remarks followed by some point to-point questions.

What was your hypothesis when setting up this study? Was it a pre planned study or were these findings during general data collection. Because with so many statistical tests you will find some significant ones.

Using correlations, the exclusion of outliers is extremely delicate, especially in such a small material. Were the exclusions done according to a protocol? Were they done before you saw the magnitude of the errors or post hoc? Please describe how this was done.

And why not reporting more clinical data about the patients than their HU?

The observations are few in your spearman plots, you exclude some outliers and still the graphs do not really impress me. And all is lost if the outliers are in.

Do not mix up correlations with causal relationship. If you conclude that malunions in patients with reduced bone mineral density require a larger extent of correction you anticipate a causal relationship not just an association.

Why would volar tilt be so sensitive (and it is only 4 degrees in the graph, close to the measurement error) to bmd? I would suspect axial compression would be more sensitive but with modern locking plates any inferior bone quality is compensated given the screws are positioned correctly underneath the subchondral bone plate.

Please report what implants were used which probably influenced the final position.

Did screw position relative the subchondral bone plate influence the final position? This is possible to measure

Volar tilt correction depends on sufficient release of the brachioradialis. Please report how the BR was managed.

Is not age a useful proxy to BMD to be used to analyze your full material with more patients to see if plate type, distance of screws to subchondral bone, handling of BR, magnitude of lengthening would correlate?

Why do you use artificial bones without any BRs and DOBs and other soft tissue components or varying BMDs that you claim make a difference?

Page 2 Abstract

When first reading your abstract I could not figure out what the paper was about. It is not immediately obvious why bone mineral density nor maximum planned distraction would cause errors and further why you involved surgery in plastic model when you have real patients. I write this now after reading the whole article and now I understand more. So you have to write it clearer for the reader. I think you should add some form of hypothesis- why do you think these parameters would matter when setting up the study.

Background

Page 3 line 15 One such factor is bone quality, particularly in osteoporotic patients. Bone settling in osteoporotic bone is a major concern

Why do you not measure this settling or subsidence? Report implant use- plate or something else? Bone settling a minor problem in locking plates.

Page 3 Line 19 Another parameter influencing the outcome is the planned maximum distraction to restore anatomical alignment, which may be limited by soft tissue tension.

Report how you dealt with the BR, and the dorsal soft tissue

Page 4 line 1 The effect of the clinical parameters on the correction accuracy can be measured directly.

You have a 4 degree VT difference according to the graph. Is that clinically important?

Page 5 line 11 All CT images (pre- and post-operative)…

Did you compare CT images (postOP) immediately after surgery or after healing (to measure subsidence, screw position etc)?

Page 5 line 21 Why specify to the lateral column? Is the maximum distance ever in the intermediate column? Or did you measure the radial/dorsal border maybe?

Page 6 line 2 What does TRU stand for?

Page 6 line 17 Outliers were identified as observations that noticeably differed from general trends.

Maybe there was no general trend? Maybe all the observations were simply random? I do not think you can continue to remove outliers until all of a sudden you find a trend? 1-2 removals from a data set of 13 are substantial.

Page 7 Line 9 Outliers again. Did you decide to remove these because of variants in surgical techniques BEFORE knowing the values of the error? That would be the only way to do such an exclusion.

Page 7 Line 20 With the two outliers included, the correlation is gone as you write. So it is a delicate decision to exclude.

Page 10 Line 8 The negative impact of low bone mineral density on the accuracy and stability of fixation of the distal radius osteotomy has been described before”

In ref 19 Ring and Jupiter write about how osteoporosis influences the radiographic outcome of a distal radius fracture, an uncontroversial topic, but do not mention the connection to radius osteotomy.

Page 10 Line 10 …the first to explore the correlation between bone mineral density

and loss of correction in VT, RI, and RL..

I think this is because the BMD is not important anymore if you use modern locking plates.

Page 10 Line 14-16 We found a strong negative correlation (rs=-0.8, p=0.001) between maximum planned distraction and bone mineral density in our cohort (Fig 3), probably a consequence of more dorsal comminution and fracture collapse in patients with osteoporosis.

A correlation is not a proof of a causal relationship. You cannot draw that conclusion. And there is a risk of mass significance with so many statistical tests.

Page 10 Line22 …leading to settling of the distal fragment on the distal screws of the plate, which primarily leads to rotation and loss in VT.

If you measure the distance of the distal screws to the subchondral bone you will know if this hypothesis is plausible. You can plot the distance vs the HU but you can also compare the distance immediately postop vs after osteotomy healing.

Page 11 line 2 Furthermore, using plates with variable screw angles could contribute to the errors as opposed to fixed-angle plates.

Interesting -Is it so? Do you have a reference for that? It is a source of technical error but in this study the surgeons are experienced. But agree- i prefer to use fixed angle plates for this.

Page 11 Line 6 To minimize the residual errors in osteoporotic patients, 3D-printed, patient-specific plates could be considered

That is a wild guess. Why would it reduce the errors? And do refs 20 and 21 really discuss the residual errors?

Page 11 line 18 The secondary aim of this study was to compare the residual errors between actual and simulated surgeries.

This part is difficult to interpret without knowing if you compared the model to the surgical result immediately postop or after healing.

Page 12 line 6 The major limitation of this study is the small cohort…

Here you list all the reasons I have raised that the results of your study maybe not are robust enough. I think they are interesting but then need to be tested in a prospective protocol and with a more homogenous material. I do not think, however, you will find any correlation between BMD and VT in the plate fixed if the screws are placed correctly.

6. PLOS authors have the option to publish the peer review history of their article (what does this mean?). If published, this will include your full peer review and any attached files.

Reviewer #1: **Yes:** Pr Marc-Olivier Gauci

Reviewer #2: **Yes:** Peter Joseph Mounsef

Reviewer #3: No

---

## [Author Response · Author response to Decision Letter 1]

18 Mar 2026

My responses to reviewers' comments are attached in the file Response to reviewers.

---

## [Decision Letter · Decision Letter 1]

16 Apr 2026

PONE-D-25-62620R1Discrepancies between in vivo and simulated correction in 3D-planned distal radius osteotomies: the influence of biological factors.PLOS One

Dear Dr. Gryska,

Thank you for submitting your manuscript to PLOS ONE. After careful consideration, we feel that it has merit but does not fully meet PLOS ONE’s publication criteria as it currently stands. Therefore, we invite you to submit a revised version of the manuscript that addresses the points raised during the review process.

We look forward to receiving your revised manuscript.

Kind regards,

Furqan A. Shah

Academic Editor

PLOS One

Journal Requirements:

Reviewers' comments:

Reviewer's Responses to Questions

**Comments to the Author**

1. If the authors have adequately addressed your comments raised in a previous round of review and you feel that this manuscript is now acceptable for publication, you may indicate that here to bypass the “Comments to the Author” section, enter your conflict of interest statement in the “Confidential to Editor” section, and submit your "Accept" recommendation.

Reviewer #2: All comments have been addressed

Reviewer #3: (No Response)

2. Is the manuscript technically sound, and do the data support the conclusions?

Reviewer #2: Yes

Reviewer #3: Yes

3. Has the statistical analysis been performed appropriately and rigorously? 

Reviewer #2: Yes

Reviewer #3: Yes

4. Have the authors made all data underlying the findings in their manuscript fully available?

Reviewer #2: Yes

Reviewer #3: Yes

5. Is the manuscript presented in an intelligible fashion and written in standard English?

Reviewer #2: Yes

Reviewer #3: Yes

6. Review Comments to the Author

Reviewer #2: (No Response)

Reviewer #3: Thank you for extensive work with the paper. Most of the text has been replaced and I had to ask for a clean sheet without changes to be able to read. But the paper has improved, is much easier to read and more focused on what you have found and is able to conclude upon.

If you see it as a limitation with the low number of patients, still do so many statistical test that you have to use bonferoni, and cannot really explain the error in VT, I would suggest you just report this as an exploratory study, show differences as 95% CI between simulation and saw bones and refrain from any statistical analyses? The study is not powered to show any differences. But it is also ok as it is with the correlations without removal of outliers.

I still have some questions as listed below below, but far less than the first time. The page and line numbers correspond to the the new clean paper without changes.

Line 8 You list technical factors such as guide-to-bone fit [6], the design of the PSSG and the resolution of the CT images used for planning as obstacles for clinical studies due to variation between patients. But why would 3D printed patient specific models make that more simple/reliable. The errors would be the same? And as you write you introduce new sources of errors like soft tissue tension bmc/d etc.

Page 6 line 5 Were the plate brand chosen because they fitted best in the planning or was that just surgeons choice?

Page 8 line 3; 83.8 HU (range 5.8–189.6). Do you need decimals here?

Page 8 line ; 4.5 to 13.7mm Same here;. was your measurment accuracy at that level? Please check the rest of the results and discussion sections if all decimals are necessary (on line 18 you even report the measurement error on a micrometer scale (0.41° and -0.29mm).

Page 11 line 1 Radial length restoration requires distraction of the distal fragment and may therefore be more sensitive to soft-tissue tension.

I think the detachment or not of the BR is super important to allow correction. Write out that it was not documented. It is ok. But the reader wants to know.

And still distance of distal screws to subchondral bone is measurable and may explain the influence of the bone density? It is mention a little later as a limitation but why do you not simply measure it ? It is only 13 patients? You have CT data and access to mathematically skilled people. And plate position- look thru the xrays- maybe that can explain the outliers?

Page 11 line 3 And why would low bone density correlate to angular error more than length errors.

I think the latter. That makes sense just as RL and planned distraction.

Page 11 line 11 line 7 when excluding outliers,

I thought you had not excluded outliers this time ?

7. PLOS authors have the option to publish the peer review history of their article (what does this mean?). If published, this will include your full peer review and any attached files.

Reviewer #2: **Yes:** Peter Joseph Mounsef

Reviewer #3: No

---

## [Author Response · Author response to Decision Letter 2]

6 May 2026

Dear Editor and Reviewers,

On behalf of all authors, we thank you for your further suggestions, which we have addressed, as shown below. In this response letter, we cite the comments in full and refer to the comments using the following notation: R3C1 denotes Reviewer 3, Comment 1, and so on. All corresponding revisions in the manuscript have been clearly indicated using this convention.

R3C1: Thank you for extensive work with the paper. Most of the text has been replaced and I had to ask for a clean sheet without changes to be able to read. But the paper has improved, is much easier tyo read and more focused on what you have found and is able to conclude upon.

R3C1 Response: Thank you for your positive assessment and such a thorough and nuanced review of our manuscript. We believe that addressing your comments further improved the manuscript.

R3C2: If you see it as a limitation with the low number of patients, still do so many statistical test that you have to use bonferoni, exclude ”outliers” and cannot really explain the error in VT, I would suggest you just report this as an exploratory study, show differences as 95% CI between simulation and saw bones and refrain from any statistical analyses? The study is not powered to show any differences.

R3C2 Response: We thank the reviewer for this helpful comment. We have strengthened the exploratory framing throughout the manuscript, including adding "Exploring discrepancies…" to the title, and have removed the Bonferroni correction and report 95% confidence intervals for each rs value alongside uncorrected p-values. All associations are explicitly framed as hypothesis-generating and are tested on all available data. We edited the Statistical analysis section (p.7) as follows:

“Spearman’s rank correlation coefficient (rs) and corresponding 95% confidence intervals were calculated for each pair of variables, as this method is robust to small sample sizes and does not assume normality. Given the exploratory nature of the study, p-values were reported alongside effect sizes for completeness. All associations should be interpreted as hypothesis-generating.”

Accordingly, we revised the results section and Fig. 3 and avoided stating that any correlation was significant (or strong or moderate, due to wide CI intervals).

R3C3: I still have some questions as listed below below, but far less than the first time. The page and line numbers correspond to the the new clean paper without changes.

Line 8 You list technical factors such as guide-to-bone fit [6], the design of the PSSG and the resolution of the CT images used for planning as obstacles for clinical studies due to variation between patients. But why would 3D printed patient specific models make that more simple/reliable. The errors would be the same? And as you write you introduce new sources of errors like soft tissue tension bmc/d etc.

R3C3 Response: We thank the reviewer for this observation and agree that the text was unclear. The advantage of 3D-printed models is not that they eliminate technical sources of error, but that they allow the same deformity to be used repeatedly under controlled conditions. For example, to compare two different guide designs without the confound of patient-specific anatomical variation. We agree that technical errors such as guide-to-bone fit and CT resolution apply equally to simulation, and we have revised the text to make this distinction explicit. We also acknowledge, as the reviewer notes, that simulation introduces its own limitations by removing biological factors such as soft-tissue tension and bone quality, which is precisely the motivation for the present study. We revised the Introduction paragraph as follows:

“Surgical accuracy may be influenced by technical factors such as guide-to-bone fit [6], the design of the PSSG [3,5], and the resolution of the CT images used for planning [7]. Evaluating the relative contribution of such factors is difficult in clinical studies because each patient presents with a unique combination of anatomy, bone quality, and deformity. This inter-patient variability makes it challenging to isolate the effect of a single technical factor, for example, two guide designs, without confounding from patient-specific differences. Patient-specific 3D-printed bone models, which have been shown to reproduce bony surface anatomy accurately [10], provide a potential solution.”

R3C4: Page 5 Line 12 Was HU extracted from the preop CT images, that is no metal artefacts.

R3C4 Response: That’s correct, the HU value was extracted from preoperative CT, without metal artifacts. We clarified this in the text:

”The distal ulna was selected because it was consistently included in all preoperative CT scans and is anatomically adjacent to the surgical site. No metal artifacts were present in the region of interest.”

R3C5: Page 6 line 5 Were the plate brand chosen because they fitted best in the planning or was that just surgeons choice?

R3C5 Response: Thank you for this question. The choice of plates was dictated by the surgeon’s preference, and the availability of STL files provided by the manufacturer. The information was added in the manuscript, under Assessment of in vivo (IV) correction accuracy section:

“The type of plate was dictated by the surgeon’s preference, the availability of STL files, and the plate fit on the bone during virtual planning. The decision to release the brachioradialis was made at the operating surgeon's discretion but was not documented consistently.”

R3C6: Page 7 line16 this method is robust to small sample sizes and does not assume normality

But then you cannot exclude outliers that does not ”fit” to the expected line.

R3C6 Response: We thank the reviewer for this comment. We agree and confirm that all 13 patients are included in every analysis, and no data points are excluded. Individual patients are described narratively in the context of the Bland-Altman analysis to aid clinical interpretation of the limits of agreement, but this does not affect the statistical results.

R3C7: Page 8 line 3; 83.8 HU (range 5.8–189.6). Do you need decimals here?

R3C8: Page 8 line ; 4.5 to 13.7mm Same here;. was your measurment accuracy at that level? Please check the rest of the results and discussion sections if all decimals are necessary (on line 18 you even report the measurement error on a micrometer scale (0.41° and -0.29mm).

R3C7 and 8 Response: Thank you for pointing this out. We reviewed the manuscript to ensure that all decimals are consistently used and necessary as follows:

Means and ranges: one decimal

Correlation coefficients: two decimals

Bland-Altman bias and limits of agreement: one decimal

Age: whole numbers

HU values: whole numbers.

R3C9: Page 11 line 1 Radial length restoration requires distraction of the distal fragment and may therefore be more sensitive to soft-tissue tension. I think the detachment or not of the BR is super important to allow correction. Write out that it was not documented. It is ok. But the reader wants to know.

R3C9 Response: We agree and thank the reviewer for raising this point. Brachioradialis (BR) release reduces deforming soft-tissue tension and may facilitate radial length restoration and volar tilt correction. Whether BR release was performed was at the operating surgeon's discretion and was not consistently documented, and could therefore not be included in the analysis. We have added the following statement in the Assessment of in vivo correction accuracy section:

”The decision to release the brachioradialis was made at the operating surgeon's discretion but was not documented consistently. ”

R3C10: And still distance of distal screws to subchondral bone is measurable and may explain the influence of the bone density? It is mention a little later as a limitation but why do you not simply measure it ? It is only 13 patients? You have CT data and access to mathematically skilled people. And plate position- look thru the xrays- maybe that can explain the outliers?

R3C10 Response: We agree that screw distance to the subchondral bone may influence the in vivo correction outcome. However, one would first need to define and validate a summary measure across a construct comprising several screws in three dimensions to obtain reliable data. Furthermore, the outcome of interest in this study is the discrepancy between in vivo and simulated corrections, not the absolute in vivo accuracy. Since the 3D-printed models do not replicate the properties of bone, including the subchondral plate, screw positioning relative to the subchondral bone cannot be meaningfully assessed or compared in the simulated setting. This variable, therefore, cannot explain IV–SIM discrepancies and falls outside the scope of the present analysis. We agree it is relevant to studies of absolute in vivo correction accuracy and have noted this in the limitations:

”A weakness of the study is that no measurements of the screw positioning relative to the subchondral plate were included, which may influence the absolute in vivo correction outcome. However, meaningfully assessing screw positioning relative to the subchondral bone in a simulated setting is not feasible because the 3D-printed models cannot replicate bone properties.”

R3C11: Page 11 line 3 And why would low bone density correlate to angular error more than length errors. I think the latter. That makes sense just as RL and planned distraction.

R3C10 Response: We thank the reviewer for this comment. On reflection, we agree that our earlier speculation linking VT errors to bone quality was not well supported by our results and have removed it. Both RL and VT discrepancies are likely influenced by soft-tissue tension acting on the distal fragment during fixation, a factor that cannot be reproduced in the simulated setting. While correlations were observed between RL discrepancies and planned maximum distraction and BMD, we note that RL and VT are not independent. The loss of VT correction will also shorten RL. Given this interdependence and the wide confidence intervals, we have been careful not to overinterpret these associations:

“The observed IV-SIM discrepancies in both RL and VT may reflect intraoperative factors not captured in the simulation. Both RL and VT restoration require distraction of the distal fragment and may therefore be sensitive to soft-tissue tension and fixation stability. The systematic difference observed in VT suggests that intraoperative factors such as bone settling due to plate offset from the bone [5] are at play in vivo but not in SIM. Importantly, however, the interquartile range of the observed VT differences remained within clinically acceptable limits in our cohort [13–16]. Correlations were observed between RL discrepancies and both planned maximum distraction and BMD, though these should be interpreted with caution, given the wide confidence intervals and the interdependence between RL and VT, where loss of VT correction will also affect RL.”

R3C11: Page 11 line 11 line 7 when excluding outliers,

Still not sure you can do like this- exclude ”outliers” that do not fit

R3C11 Response. We thank the reviewer for this observation. To clarify, no outliers are excluded from any analysis. The original phrasing was misleading, as it referred to the visual representation in the box-and-whisker plot, where individual points outside the interquartile range are displayed as separate markers by convention, not excluded from the statistics. We have removed the ambiguous phrasing and replaced it with:

”Importantly, the interquartile range of the observed VT differences remained within clinically acceptable limits in our cohort [13–16].”

---

## [Editor Report · Decision Letter 2]

7 May 2026

Exploring discrepancies between in vivo and simulated correction in 3D-planned distal radius osteotomies: the influence of biological factors.

PONE-D-25-62620R2

Dear Dr. Gryska,

We’re pleased to inform you that your manuscript has been judged scientifically suitable for publication and will be formally accepted for publication once it meets all outstanding technical requirements.

Kind regards,

Furqan A. Shah

Academic Editor

PLOS One
---

## [Editor Report · Acceptance letter]

PONE-D-25-62620R2

PLOS One

Dear Dr. Gryska,

I'm pleased to inform you that your manuscript has been deemed suitable for publication in PLOS One. Congratulations! Your manuscript is now being handed over to our production team.

Kind regards,

on behalf of

Dr. Furqan A. Shah

Academic Editor

PLOS One